# Genetic Structure across Isolated Virginia Populations of the Endangered Candy Darter (*Etheostoma osburni*)

**Kathryn E. McBaine** [1,†], **Paul L. Angermeier** [1,2] and **Eric M. Hallerman** [1,*]

1 Department of Fish and Wildlife Conservation, Virginia Polytechnic Institute and State University, Blacksburg, VA 24061, USA; katimc9@vt.edu (K.E.M.); biota@vt.edu (P.L.A.)
2 U.S. Geological Survey, Virginia Cooperative Fish and Wildlife Research Unit, Department of Fish and Wildlife Conservation, Virginia Polytechnic Institute and State University, Blacksburg, VA 24061, USA
* Correspondence: ehallerm@vt.edu; Tel.: +1-540-553-5873
† Current Affiliation: Idaho Department of Fish and Game, 1414 E. Locust Lane, Nampa, ID 83686, USA.

**Abstract:** Candy darter *Etheostoma osburni*, a federally endangered non-game fish, has been extirpated from most of its historic range in Virginia and now occurs in four isolated populations in the New River drainage. Understanding of population genetic structure will provide insights into the recent natural history of the species and can inform conservation management. Our objectives were to: characterize population genetic structure, estimate and compare effective population sizes ($N_e$), and use this information to infer recent population history. Variation at mitochondrial cytochrome *b* sequences among 150 individuals showed 10 haplotypes separated by 1–14 mutational steps, some shared and some unique to particular populations. Variation at 12 microsatellite loci among 171 individuals showed lower variation in Dismal Creek than in other populations. All populations showed evidence of having experienced a genetic bottleneck and were highly differentiated from one another based on both types of DNA markers. Population genetic structure was related to stream position in regard to the New River, suggesting that populations were once connected. $N_e$ estimates for all populations were less than the 500 recommended to maintain evolutionary potential, but most estimates were greater than the 100 needed for use as source populations. Our findings indicate that habitat management to allow expansion of populations, and translocations to exchange genetic material among populations, may be effective tactics to promote conservation of candy darter in Virginia.

**Keywords:** conservation genetics; genetic drift; genetically effective migration; population genetics; population viability; translocation

**Key Contribution:** Our results can contribute to conservation planning for this imperiled species. Our results suggest that all populations may benefit from augmentations to overcome adverse effects of genetic drift and inbreeding and to safeguard the historical genetic variation of the species. Three of the populations may be viable source populations for translocations.

## 1. Introduction

Declining species often exhibit a shrinking spatial distribution; as habitat becomes less hospitable, populations become smaller and more isolated. Such populations suffer decreasing genetically effective immigration and increasing genetic drift, giving rise to a genetic signature of isolation. That is, small isolated populations with little or no gene flow are vulnerable to stochastic events that cause demographic bottlenecks, resulting in loss of potentially adaptive alleles [1]. Additionally, small populations are at greater risk of inbreeding depression due to increased homozygosity for deleterious recessive alleles, which could drive the population into further decline or extinction [2,3]. Population genetic studies can reveal the genetic signature of isolation, giving insights into the recent demographic history of a species, thereby informing conservation management. Managers often

consider reintroductions or translocations to increase the long-term viability of isolated populations [4,5]. An understanding of population genetic structure is needed to address which populations may serve as a source and which populations may need augmentation with new genetic variation via stocking.

Darters (Family Percidae: Subfamily Etheostomatinae) are among the most imperiled freshwater fishes in North America [6]. Habitat degradation, ecological specialization, and naturally restricted range are major factors associated with their imperilment [7]. The candy darter *Etheostoma osburni* (Figure 1), which shares many traits with other darters, is a small, riffle-dwelling, imperiled non-game fish endemic to the New River drainage in Virginia and West Virginia. It is narrowly restricted to medium-sized streams with cold-cool temperatures (mean maximum summer temperatures of 27.8 °C in occupied sites), high-velocity riffles, and silt-free substrates [8,9]. Its abundance, age structure, and life history in the four Virginia streams in which it occurs recently have been described. The species has greatest abundance in Stony and Cripple creeks [9]. Individuals can live to at least age-5, based on otolith readings. Females can mature at age-2, then reproduce in four consecutive years with multiple males each year; parentage analysis indicates that both sexes mate with multiple partners [10].

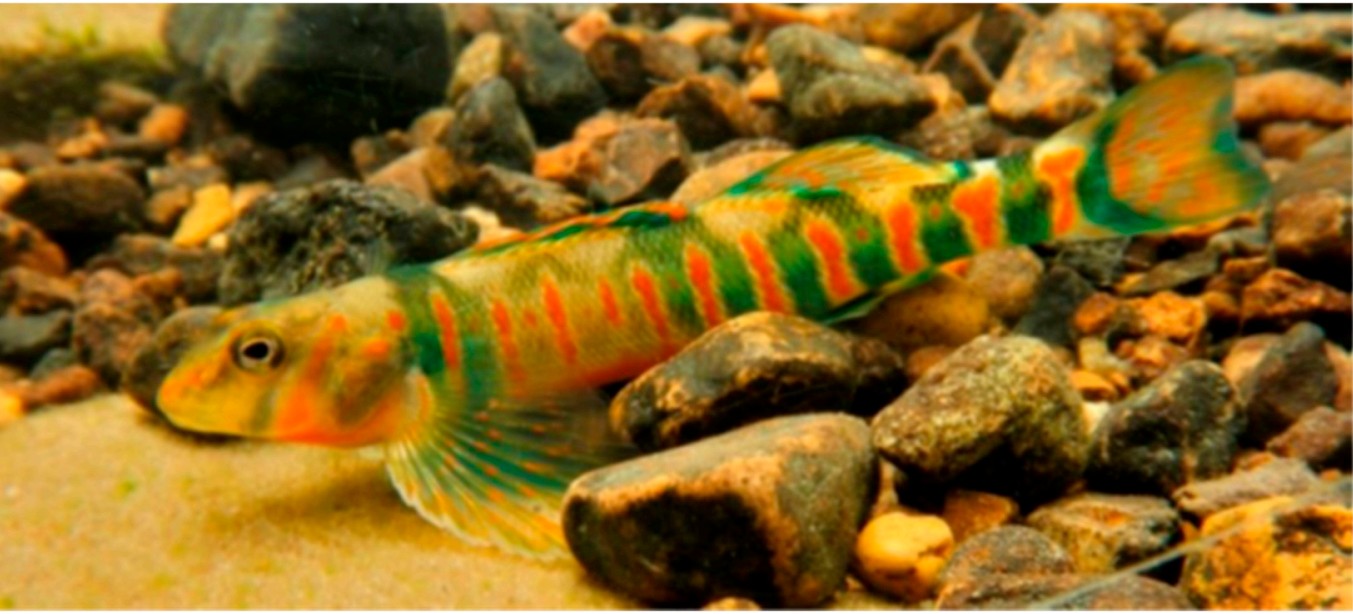

**Figure 1.** Male candy darter, *Etheostoma osburni* (photo K. McBaine).

Historically, the candy darter occurred at 35 locations distributed across the Appalachian Plateau and the Valley and Ridge physiographic provinces. Populations in the Valley and Ridge were more fragmented than those in the Appalachian Plateau, and there are different morphs in the respective regions [11]. Candy darter has been extirpated from almost half of its historical range, i.e., from 17 of 35 known locations as a result of habitat degradation from unregulated land-use practices [11] (Figure S1). Recently, hybridization with the introduced congener variegate darter *E. variatum* has become a threat to populations downstream of Bluestone Dam in West Virginia [12]. Variegate darters have expanded within the range of candy darter; where they co-occur, the two species will mate, resulting in introgressive hybridization of the endemic candy darter population and eventually in complete replacement by variegate darters or hybrids. In contrast, above Bluestone Reservoir, habitat degradation (increased sedimentation, warming water temperatures, and habitat fragmentation) and catastrophic events (e.g., toxic spills) are the greatest threats to the four isolated populations of candy darter remaining in Virginia [11]. Because historic sites of occurrence were near to one another, there may have been recent gene flow among populations in neighboring streams. However, the mouths of streams containing the four

known extant populations in Virginia are separated by >10 river kilometers (rkm) of the mainstem New River, which may be unsuitable habitat [8], making contemporary dispersal among streams unlikely. The decline of candy darter has been such that the U.S. Fish and Wildlife Service [13] listed it as an endangered species and designated critical habitat [14].

Genetic and demographic augmentation are commonly used to mitigate adverse effects of genetic drift and inbreeding on isolated fish populations [4]. In 1996, 30 adult candy darters were translocated from Stony Creek to Dismal Creek, as it was presumed that the population in Dismal Creek had been extirpated [15]. Propagation and augmentation actions have been taken in West Virginia, and further actions have been discussed in conservation planning meetings involving the U.S. Fish and Wildlife Service, U.S. Forest Service, Virginia Department of Wildlife Resources, West Virginia Department of Natural Resources, and other agencies that manage candy darter populations or whose actions may affect the species. Achieving successful augmentation requires species-specific knowledge of population genetic structure, including genetic variation, effective population size ($N_e$), and demographic history of populations. Such knowledge is currently lacking for candy darter. Against this background, we developed datasets useful for characterizing the genetic and demographic dynamics of candy darter populations in Virginia. In this first report of the population genetics of the species, our objectives were to: 1) describe population genetic structure, 2) estimate and compare effective population sizes of the four extant populations in Virginia, and 3) infer recent population history. Enhanced understanding of the viability and dynamics of extant populations can inform conservation planning for the species.

## 2. Materials and Methods

### 2.1. Study Site

We sampled four streams with extant populations of candy darter in the middle and upper New River drainage, Virginia (Table S1, Figure 2). Candy darters were sampled throughout their distribution within each occupied stream to represent longitudinal gene flow and diversity. Over 50 rkm separate these populations, and approximately 186 rkm separate the most disjunct populations. Cripple Creek and Stony Creek are tributaries to the New River. Cripple Creek represents the most southern occurrence of candy darter, and fluvial distance and Claytor Dam on New River isolated its population. Candy darter occupies the lower 8 km of Cripple Creek and lower 18.8 km of Stony Creek. However, the lower 1.5 km of Stony Creek flows underground, leaving a dry streambed during the fall and winter months. McBaine and Hallerman [9] regarded these two populations as the largest of the Virginia populations based on catch-per-unit-effort data and abundance estimates. Laurel Creek is a small tributary to Wolf Creek, which then flows approximately 37 km to the New River. In Laurel Creek, a milldam separates the lower 4.25 km of candy darter occupancy from the upstream portion, where Dunn and Angermeier [16] regarded them as absent. Dismal Creek is a small tributary to Kimberling Creek which empties into Walker Creek, a tributary of the New River. Dismal Creek has several natural falls that may limit dispersal by candy darter, as well as introduced species. Candy darter occupy 4.2 km of Dismal Creek, representing the smallest of the Virginia populations.

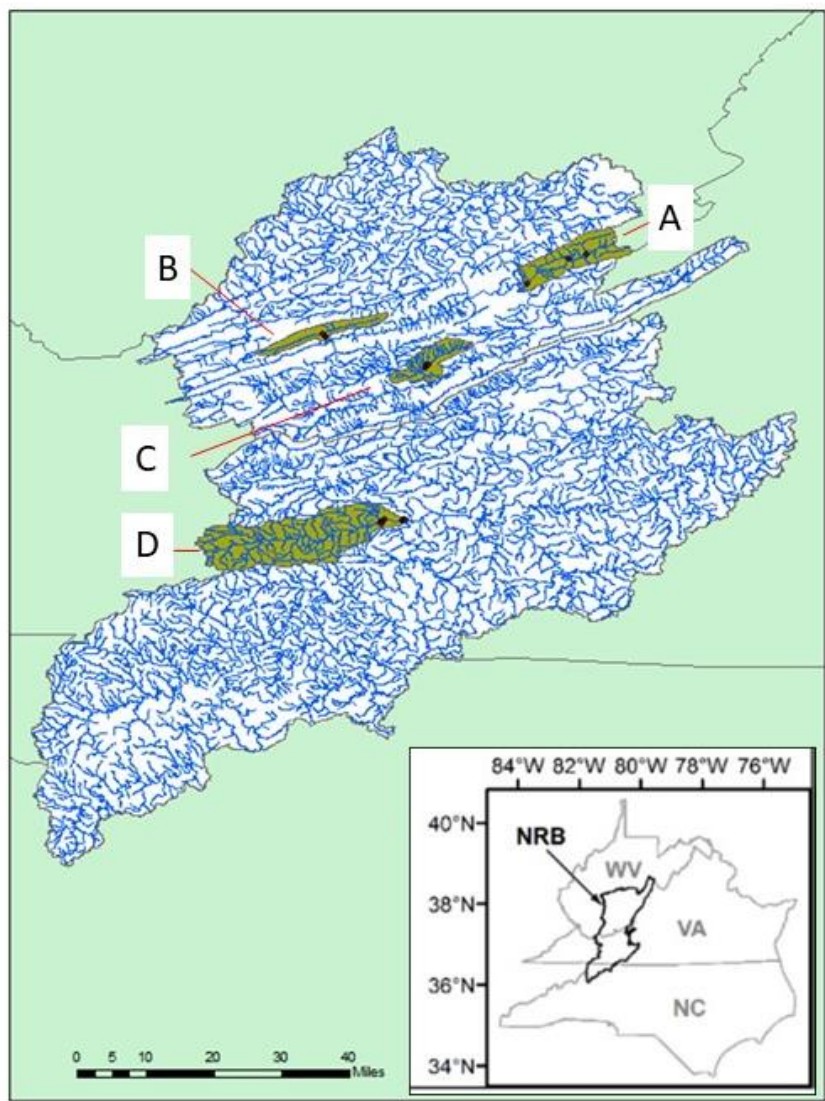

**Figure 2.** Middle and upper New River drainage (including West Virginia—WV, Virginia—VA, and North Carolina—NC), including sites (A—Stony Creek, B—Laurel Creek, C—Dismal Creek, and D—Cripple Creek) for study of population genetics of candy darter *Etheostoma osburni*. Sampling locations are shown as dots. Inset map: Location of entire New River basin (NRB), including lower, middle, and upper portions, within eastern North America.

### 2.2. Field Methods

We conducted fish surveys during May-September of 2016–2018. We sampled candy darters using pulsed direct current (DC) from a backpack electrofisher (Smith-Root LR-24). A 1.5 × 3-m weighted seine with 5-mm mesh was held by two crew members, as a third person electrofished, working downstream while disturbing the substrate, thereby allowing the stream flow to carry stunned fish into the seine. Upon capture of a candy darter, we collected length (standard and total) and sex data. We anesthetized candy darters in an immersion solution of AQUI-SE (AQUI-S New Zealand, Ltd., Melling, Lower Hutt, New Zealand) and stream water. Fin clips were taken from the lower lobe of the caudal fin for every individual, then air-dried in a scale envelope labeled with a unique individual alphanumeric code corresponding to the stream and site. After processing, we placed fish in a recovery tank of stream water until normal behaviors resumed, then released them. This work was carried out under the auspices of Virginia Tech

Institutional Animal Care and Use Committee Protocol 16-095, first approved 29 September 2016.

### 2.3. Mitochondrial DNA

We used mitochondrial DNA to assess population structure and historical maternal gene flow patterns to identify possible source populations for translocations or reintroductions. We amplified a 965-bp fragment of the mitochondrial cytochrome-*b* gene using primers that we developed from the DNA sequence reported in GenBank accession number HQ128185 [17]: forward—5′–GTGACTTGAAAAACCACCGTTG–3′ and reverse—5′–CAACGATCTCCGGTTTACAAGAC–3′. We performed PCR amplifications in a final volume of 25 µL which contained 7.0 µL of DNA extract, 9.87 µL nanopure water, 5.0 µL 5× buffer (GoTaq Flexi buffer, Promega, Madison, WI, USA), 1.0 µL 25 mM MgCl$_2$ (Promega), 0.50 µL dNTPs (Promega), 0.75 µL of each primer, and 0.125 units/µL of DNA polymerase (GoTaq, Promega). The PCR protocol consisted of an initial denaturation at 95 °C for 2 min; 42 cycles of: 94 °C denaturation 1 min, 52 °C annealing for 1 min, and 72 °C extension for 2 min; and a 5-min extension at 72 °C. We used an aliquot of the PCR product for confirmation of amplification of DNA by UV visualization of an ethidium bromide-stained band in a 2% agarose gel. We sent PCR products showing amplification to the Fralin Life Sciences Institute (Blacksburg, VA, USA) for DNA sequencing (3730XL DNA Analyzer, ABI, Waltham, MS, USA). We assembled raw DNA sequences (Geneious Prime 2019.2, Geneious, Auckland, New Zealand) and aligned them (GeneStudio Professional Edition Version 2.2, GeneStudio, Inc., http://www.genestudio.com/).

We used mitochondrial DNA sequences from three Kanawha darters (*E. kanawhae*; GenBank accession numbers HQ128150.1, AY964689.1, and AF411381.1) as an outgroup for phylogenetic analysis, as they represent a closely related, but distinct species. We determined relations among haplotypes of candy darters and Kanawha darters using DnaSP version 6.12 [18]. In addition, we assessed pairwise mitochondrial DNA sequence mismatch distributions within populations as an indication of historical demography conducted using DnaSP. We constructed the mitochondrial cytochrome-*b* sequence haplotype network using TCS network [19] in PopART 1.7 [20].

We conducted analysis of molecular variance (AMOVA) among individuals within populations and among populations using Arlequin 3.5.2.2 [21]. We calculated the population differentiation metric $F_{ST}$ using Arlequin, and assessed the significance of its departure from zero using a randomization algorithm with 10,000 iterations.

We used the MrModeltest [22] software applied within the PAUP software package [23] to determine the most appropriate mutation model for characterizing phylogenetic relationships. The best-fit model was the Hasegawa-Kishino-Yano 85 (HKY85, [24]) model, which allows transitions and transversions to have different rates. We conducted Bayesian phylogenetic analyses with MrBayes 3.1.2 [25]. We set the Markov chain Monte Carlo (MCMC) process to conduct four chains and run for 660,000 cycles with an average standard deviation < 0.01 with other parameters set to default. We used FigTree v 1.4.4 [26] to visualize the final phylogenetic tree.

### 2.4. Microsatellite DNA

We used microsatellite DNA to assess population structure and effective population size to identify appropriate source populations for either translocations or reintroductions. We screened 13 nuclear microsatellite loci for genetic variation using primer pairs developed by Switzer et al. [27] for this species and adapted their PCR protocols. We amplified DNA using three multiplex PCR reactions (1: *EosD116, EosD107, EosC124, EosC6*; 2: *EosC208, EosC207, EosC112, EosC117*; 3: *EosD10, EosC3, EosC2, EosD108, EosD11*). We performed PCR amplifications in a final volume of 10 µL, which contained 2.0 µL of DNA extract, 2.0 µL nanopure water, 2.0 µL 5× buffer (GoTaq Flexi, Promega), 1.75 µL 25 mM MgCl$_2$ (Promega), 1.15 µL 2.5 µM dNTPs (Promega), 0.5 µL of each primer, and 0.1 units/µL of DNA polymerase (GoTaq, Promega). The PCR protocol consisted of an initial denaturation at 95 °C for 15 min; 25 cycles of: 94 °C denaturation 30 s, 57 °C annealing for 90 s, and 72 °C extension for 1 min; and a 30-min extension at 60 °C. We used an aliquot of the PCR product for confirmation of amplification of DNA in a 2% agarose gel. We sent PCR

products to the Cornell University Core Laboratory (Ithaca, NY, USA) for fragment-size analysis using an ABI 3730XL DNA Analyzer.

We used MicroChecker version 2.2.3 [28] to test for segregation of null alleles, large-allele drop out, and replication stutter with 1000 randomizations and a Bonferroni-corrected significance level. We used Arlequin v3.5.2.2 [21] to test for departures from Hardy-Weinberg equilibrium and linkage disequilibrium for populations in all four streams. Hardy-Weinberg tests had 105 Markov Chain Monte Carlo (MCMC) iterations, following a burn-in of 103 iterations. Linkage tests had 105 randomizations. We estimated mean number of alleles per locus, allelic richness, expected heterozygosity ($H_E$), and observed heterozygosity ($H_O$) for populations in each stream using Arlequin. We calculated the Garza-Williamson [29] $m$ index, which indicates a bottleneck at values < 0.70 [29], using Arlequin.

We used Arlequin to calculate $F_{ST}$ to quantify population differentiation and to conduct analysis of molecular variance (AMOVA) and used the same significance testing scheme (10,000+ permutations) as the mtDNA $F_{ST}$ comparisons. Using the criterion of at least 10% exchange [30], we considered groups spawning at different sites demographically isolated if they exchanged fewer than 10% of adults, which in this case corresponds to genetic differentiation ($F_{ST}$) of 0.021 under a classical Wright–Fisher island model of migration-drift equilibrium. We estimated inbreeding coefficients while accounting for segregation of null alleles using INEST [31].

Additionally, we analyzed population structure using Bayesian spatial clustering models to define multilocus genotypic clusters and to assign individual multilocus microsatellite genotypes into those clusters using STRUCTURE 2.3.4 [32]. We evaluated population structure for $K$ = 1–8 clusters to account for possible structuring within and between streams. All models allowed for admixture and correlation of allele frequencies among clusters and parameter space using 106 MCMC iterations, following a burn-in of 105 iterations. We repeated runs 10 times for each value of $K$. We selected the replicate run with the lowest Bayesian deviance (=$-2$ log likelihood) as the most likely estimate of that $K$-value [33]. We assessed the best-supported value of $K$ (empirically defined populations) using the Evanno et al. [34] method and the highest log-likelihood of the data given $K$, an output metric provided by the STRUCTURE software.

We estimated effective population size using NeEstimator v 2.1 [35] with the random mating model and parametric confidence intervals. Because including rare alleles in linkage disequilibrium analysis can upwardly bias estimates of effective population size [3], we did not include allele frequencies below 0.02 in the analysis. We selected the random mating model as opposed to the lifetime monogamy model as candy darter exhibits a polygamous mating system [10]. Since we combined multiple year-classes, the estimated $N_e$ values represent something between $N_b$, the number of breeders in any one year, and the effective population size per generation [36].

## 3. Results

### 3.1. Mitochondrial DNA

We amplified 150 mitochondrial cyt-*b* sequences 965 bp in length (Cripple Creek *n* = 36, Dismal Creek *n* = 24, Laurel Creek *n* = 42, Stony Creek *n* = 48). Fourteen variable sites defined 10 cyt *b* haplotypes (Table 1). Populations in the respective creeks had distinct haplotypes separated by 1 to 14 mutational steps. Three haplotypes were observed in Cripple Creek, one in Dismal Creek, and two each in Laurel and Stony creeks. The Laurel Creek population shared two haplotypes, one with the Cripple Creek and the other with the Dismal Creek population (Figure 3). Three unique/private haplotypes were observed in Cripple Creek, one in Dismal Creek, two in Laurel Creek, one in Stony Creek, and three in the outgroup *E. kanawhae*. The *E. kanawhae* haplotypes occurred among the *E. osburni* haplotypes, a surprising result to which we return in the Discussion.

**Table 1.** Variable nucleotide sites at the mitochondrial cytochrome-*b* gene and counts of haplotype (Hap_) occurrence in candy darters (*Etheostoma osburni*) from four streams. *N* is the number of individuals with the respective haplotype; "." represents the same nucleotide as observed in Hap_1.

| Haplotypes | N | Variable Sites | | | | | | | | | | | | | | Counts | | | |
| | | 1 6 7 | 3 3 6 | 3 7 7 | 4 4 3 | 4 5 2 | 5 9 0 | 6 7 7 | 8 0 0 | 8 4 2 | 8 4 8 | 8 8 4 | 9 2 6 | 9 3 4 | 9 3 6 | Dismal Creek | Cripple Creek | Laurel Creek | Stony Creek |
|---|---|---|---|---|---|---|---|---|---|---|---|---|---|---|---|---|---|---|---|
| Hap_1 | 33 | T | T | G | T | G | A | C | T | A | T | T | C | T | A | 0 | 18 | 15 | 0 |
| Hap_2 | 47 | . | . | . | . | . | G | . | C | . | C | . | . | . | . | 0 | 0 | 0 | 47 |
| Hap_3 | 1 | . | . | . | . | . | G | . | C | . | C | . | . | A | . | 0 | 0 | 0 | 1 |
| Hap_4 | 14 | C | . | . | C | A | G | T | C | . | C | . | . | . | G | 0 | 14 | 0 | 0 |
| Hap_5 | 16 | C | . | . | C | A | G | T | C | . | C | . | . | . | . | 13 | 0 | 3 | 0 |
| Hap_6 | 11 | C | C | . | C | A | G | T | C | . | C | . | . | . | . | 11 | 0 | 0 | 0 |
| Hap_7 | 3 | C | . | . | C | A | G | T | C | G | C | . | . | . | . | 0 | 3 | 0 | 0 |
| Hap_8 | 18 | . | . | . | . | . | . | . | C | . | . | C | . | . | . | 0 | 0 | 18 | 0 |
| Hap_9 | 6 | . | . | . | . | . | . | . | C | . | C | . | T | . | . | 0 | 0 | 6 | 0 |
| Hap_10 | 1 | . | . | A | . | . | . | . | C | . | C | . | . | . | . | 0 | 1 | 0 | 0 |

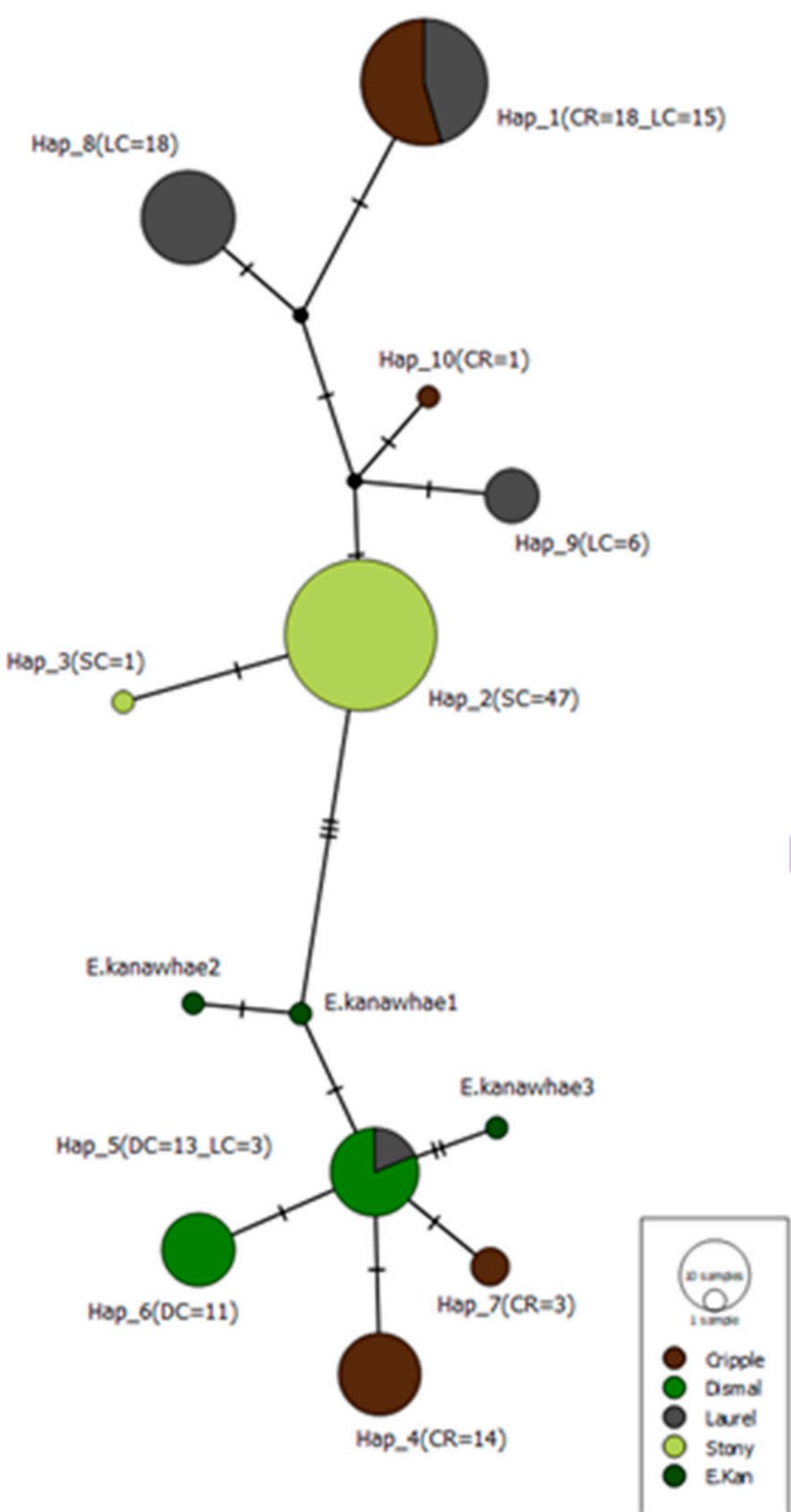

**Figure 3.** Mitochondrial cyt *b* DNA sequence haplotype network for candy darter *Etheostoma osburni* populations in Cripple (CR), Dismal (DC), Laurel (LC), and Stony (SC) creeks, with three Kanawha darter (*E. kanawhae* [E.Kan]) haplotypes shown as an outgroup. Sizes of circles are roughly proportional to the number of fish sampled within the respective haplotypes. Numbers in parentheses are the number of individuals from the respective population present for each haplotype.

DNA sequence mismatch distributions provide insight into the demographic history of populations. If only mutation and random genetic drift had been at issue, observed frequencies of mismatch (shown in red in Figure S3) would match the expected frequencies (shown in green). Observed DNA sequence mismatch distributions in Dismal and Stony creeks (Figure S3b,d) match reasonably well, suggesting population genetic isolation and recent genetic bottlenecks. However, secondary peaks for the Cripple and Laurel creek populations (Figure S3a,c) suggest they have had recent secondary contact with other differentiated populations.

The partitioning of molecular variance of mitochondrial cyt-*b* sequences into within- and among-population components using AMOVA (Table 2A) showed a high level of genetic variance (59%) among populations. Additionally, $F_{ST}$ values for mitochondrial cyt-*b* sequences among all population pairs (Table 3) were high, ranging from 0.27–0.95, with the greatest differentiation between populations in Stony and Dismal creeks (0.95). All $F_{ST}$ estimates were statistically significant at an adjusted $\alpha = 0.083$. The phylogenetic tree and haplotype network revealed the same patterns among population clusters. The consensus phylogenetic tree had bootstrap support above 99% for well-supported clades. The maximum likelihood analysis produced a log likelihood of $-1502.23$. The Bayesian analysis resulted in a 99% credible set of 9803 trees of 9902 trees sampled. Two of the three Kanawha darter sequences were outside the candy darter clusters, as expected for an outgroup (Figure S2). However, the third sequence was clustered with the Dismal and Laurel creek populations. The phylogenetic tree and haplotype network grouped Laurel and Cripple creek haplotypes on a distinct branch, and Dismal and Laurel creek haplotypes on another distinct branch. However, Stony Creek haplotypes did not share a branch with haplotypes from any other creek, and Dismal Creek haplotypes did not share a branch with Cripple Creek haplotypes.

**Table 2.** Analysis of molecular variance (AMOVA) results for: **A.** Mitochondrial cytochrome-*b* sequences, and **B.** Microsatellite DNA markers for four candy darter (*Etheostoma osburni*) populations in Virginia. *df* = degrees of freedom. $V_a$ = variance among populations; $V_b$ = Variance among individuals within populations; $V_c$ = Variance within individuals.

| A | | | | |
|---|---|---|---|---|
| **Source of Variation** | *df* | **Sum of Squares** | **Variance Components** | **% of Variation** |
| Among populations | 3 | 140.61 | 1.25 $V_a$ | 58.7 |
| Among individuals within populations | 146 | 128.15 | 0.88 $V_b$ | 41.3 |
| Total | 149 | 268.76 | 2.13 | |
| B | | | | |
| **Source of Variation** | *df* | **Sum of Squares** | **Variance Components** | **% of Variation** |
| Among populations | 3 | 308.00 | 1.17 $V_a$ | 24.0 |
| Among individuals within populations | 167 | 714.36 | 0.56 $V_b$ | 11.5 |
| Within individuals | 171 | 539.50 | 3.15 $V_c$ | 64.5 |
| Total | 341 | 1561.86 | 4.89 | |

Fixation Index $F_{ST} = 0.59$, Significance tests (10,100 permutations), $V_a$ and $F_{ST}$: $P$ (random value > observed value) = 0.00000.

**Table 3.** Pairwise fixation index ($F_{ST}$) values for mitochondrial cytochrome-*b* sequences (above diagonal) and for microsatellites (below diagonal) for four candy darter (*Etheostoma osburni*) populations in Virginia. All non-diagonal values were significantly different ($p < 0.05$) from zero.

| Stream | Cripple Creek | Dismal Creek | Laurel Creek | Stony Creek |
|---|---|---|---|---|
| Cripple Creek | 0 | 0.44 | 0.27 | 0.49 |
| Dismal Creek | 0.28 | 0 | 0.77 | 0.95 |
| Laurel Creek | 0.19 | 0.33 | 0 | 0.63 |
| Stony Creek | 0.17 | 0.31 | 0.25 | 0 |

### 3.2. Microsatellite DNA

Based on an analysis of 13 microsatellite loci for individuals of each population (Cripple Creek *n* = 49 individuals, Dismal Creek *n* = 25, Laurel Creek *n* = 47, Stony Creek *n* = 50), genetic diversity was low for all four populations. The mean number of microsatellite alleles per locus was similar among populations, except the mean was lower in Dismal Creek (Table 4, Table S2). All populations were fixed for a single allele at *Eos-C2*, so we removed that locus from further analysis. Two of the 12 remaining loci showed evidence of segregating null alleles. Our analyses of the reduced dataset (10 loci) and the full dataset (12 loci) produced similar results. The Laurel Creek population had the greatest mean number of alleles per locus and the Stony Creek population had the highest allelic richness. Dismal Creek had the lowest mean number of alleles and lowest allelic richness. Based on a Bonferroni correction ($\alpha = 0.004$), we found departures from Hardy-Weinberg equilibrium (HWE) in each population at multiple loci (Cripple *n* = 2, Dismal *n* = 8, Laurel *n* = 4, Stony *n* = 2). Observed heterozygosity was less than expected for most loci in each population (Table 4 and Table S2); the Cripple Creek population had the greatest mean observed heterozygosity. The inbreeding coefficient ($F_{IS}$) was relatively high for all populations (>0.09; Table 5) but greatest in Dismal Creek ($F_{IS} = 0.45$). However, INEST results accounting for segregation of null alleles suggested lower levels of inbreeding for all populations (Table 5). The INEST results were most compatible with small population sizes, family structure, the occurrence of inbreeding, and other deviations from HWE. The m-ratios for all populations were <0.11, much lower than the 0.68 criterion suggested by Garza and Williamson [29] for identifying recent reductions in population size. These low values indicate a recent genetic bottleneck for each stream-specific population (Table 4).

**Table 4.** Genetic diversity metrics for four candy darter (*Etheostoma osburni*) populations from Virginia across 12 microsatellite loci: *N* = number of fish sampled, *A* = mean number of alleles per locus, $A_r$ = allelic richness, $H_o$ = mean observed heterozygosity, $H_e$ = mean expected heterozygosity, Allelic Range = mean difference between sizes of largest and smallest alleles at a particular locus, *M* = ratio of number of alleles observed at a locus to number of alleles possible between the largest and smallest alleles (Garza and Williamson 2001, [29]).

| Stream | *N* | *A* | $A_r$ | $H_o$ | $H_e$ | Allelic Range | *M* |
|---|---|---|---|---|---|---|---|
| Cripple Creek | 98 | 6.67 | 52.33 | 0.64 | 0.70 | 161.00 | 0.11 |
| Dismal Creek | 50 | 3.92 | 26.96 | 0.25 | 0.46 | 247.33 | 0.02 |
| Laurel Creek | 94 | 7.00 | 50.50 | 0.56 | 0.63 | 188.67 | 0.08 |
| Stony Creek | 94 | 7.00 | 55.19 | 0.56 | 0.63 | 188.67 | 0.08 |

**Table 5.** Inbreeding coefficients for four populations of candy darter (*Etheostoma osburni*) in Virginia, with $F_{IS}$ metrics from Arlequin [21] and *F*i mean from INEST [31]. Results from Arlequin include testing the probability that a random $F_{IS}$ would be greater than the observed $F_{IS}$. $F_i$ mean = mean inbreeding coefficient for all loci; CI = confidence interval.

| Stream | Arlequin | | INEST | |
|---|---|---|---|---|
| | $F_{IS}$ | *P* (Random $F_{IS} \geq$ Observed $F_{IS}$) | $F_i$ Mean | 95% CI |
| Cripple Creek | 0.09 | <0.001 | 0.02 | 0.0–0.05 |
| Dismal Creek | 0.45 | <0.001 | 0.06 | 0.0–0.15 |
| Laurel Creek | 0.14 | <0.001 | 0.02 | 0.0–0.06 |
| Stony Creek | 0.12 | <0.001 | 0.03 | 0.01–0.06 |

We estimated high levels of differentiation for all pairwise comparisons of microsatellite loci between populations (Table 2B). The Laurel Creek and Dismal Creek populations showed the greatest differentiation, while Stony Creek and Cripple Creek populations were the least differentiated. Following Palsboll et al. [37], we considered populations demographically independent if they exchanged fewer than 10% migrants per generation. We estimated the critical $F_{ST}$ value corresponding to the 10% threshold using the average $N_e$ of 144 across all populations (see below), yielding $F_{STcritical}$ = 0.02. The observed $F_{ST}$ being greater than this critical value, all Virginia candy darter populations are demographically independent.

The AMOVA partitioned genetic variance into three components: within individuals, among individuals within populations, and among populations. Variation within individuals was the greatest component (65%), with 11% of variance among individuals within populations and 24% of variance among populations (Table 2B).

The best-supported Bayesian model of population genetic structure was $K = 4$ multilocus genotypic clusters (Figure S4, Table S3). STRUCTURE clustered individuals within the respective streams as separate populations (Figure 4). At higher $K$-values, these clusters broke down further, but there was no indication of within-stream structuring across sites.

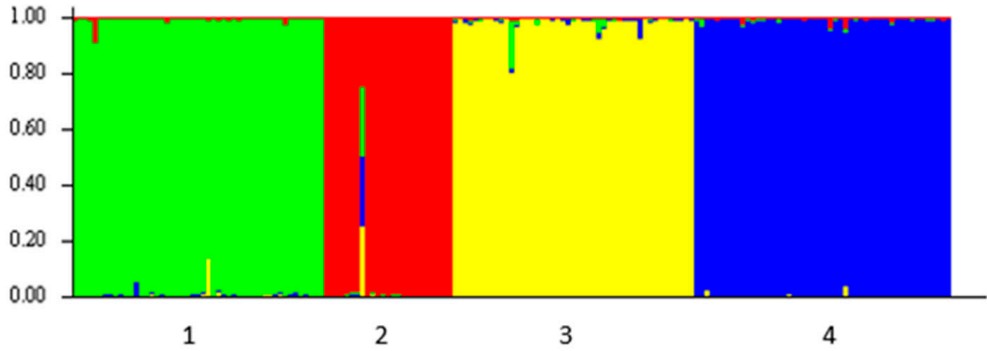

**Figure 4.** STRUCTURE plot representing population genetic structure of candy darter (*Etheostoma osburni*) populations in Virginia for $K = 4$ clusters of multilocus microsatellite DNA genotype data. Populations are from (1) Cripple Creek (green), (2) Dismal Creek (red), (3) Laurel Creek (yellow), and 4) Stony Creek (blue). The *Y*-axis is the proportion of each individual (336 total) assigned to a particular cluster (population).

Based on the sample size for each stream, we considered $N_e$ at allele frequencies of 0.02 as conservative estimates. Noting further that because we combined data for multiple year-classes, the estimated $N_e$ values represent something between $N_b$, the number of breeders in any one year, and the effective population size per generation, conservative $N_e$ estimates ranged from 39 in Dismal Creek to 223 in Cripple Creek (Table 6). Except for Cripple and Laurel creeks, nearly all estimates had undefined upper bounds, reflecting the imprecision of our $N_e$ estimates.

**Table 6.** Effective population size ($N_e$) estimates, with confidence intervals (CIs), for four populations of candy darter (*Etheostoma osburni*) in Virginia. Parametric and Jackknife CIs represent different methods for estimating the precision of $N_e$ estimates [35]. *n* = number of individuals sampled. All alleles included in the analysis had frequencies of at least 0.02.

| Stream | *n* | $N_e$ | Parametric CI | Jackknife CI |
|---|---|---|---|---|
| Cripple Creek | 49 | 172.6 | 85.3–1682 | 63.5–∞ |
| Dismal Creek | 25 | 39 | 10.3–∞ | 7.9–∞ |
| Laurel Creek | 47 | 136.1 | 73.9–520.2 | 57.7–∞ |
| Stony Creek | 50 | 192.7 | 82.3–∞ | 54–∞ |

## 4. Discussion

### 4.1. Historical Demography and Genetics of Candy Darter in the Upper New River

Analyses of the phylogenetics and population genetics of candy darter above Bluestone Dam provided insights into the species' historic and recent population dynamics. Interpretation of results from the mitochondrial and microsatellite DNA screenings support the view that candy darter historically occurred throughout the upper New River drainage and may have dispersed regularly between streams, thereby exhibiting metapopulation dynamics. However, human-mediated habitat and landscape changes apparently have led to extirpations of spatially intervening populations, and inhospitable dispersal routes (possibly including the New River) led to demographic and genetic isolation of the four extant populations in Virginia. These populations subsequently experienced random genetic drift and have lost considerable genetic diversity as evidenced by the respective populations not exhibiting all mitochondrial haplotypes or microsatellite alleles observed among all populations. As noted below, a signature of population genetic structuring was notable among extant populations.

Mitochondrial cytochrome-*b* haplotype frequencies showed clear divergence between all populations (Table 1). The large number of haplotypes and the lack of spatial structure within the haplotype network (Figure 3) suggest that there was a large, relatively recent ancestral candy darter population within the New River watershed that contained most or all of these haplotypes. However, it is likely that random genetic drift has occurred subsequently, leading to loss of haplotypes within each of the respective populations, as evidenced by the small numbers of haplotypes within contemporary populations (range, 2–4) and of shared haplotypes (2). All pairwise $F_{ST}$ values among populations in the respective streams (Table 3) reflect high levels of differentiation in mitochondrial cyt-*b* sequences. This pattern suggests long-term isolation, declining population sizes, and loss of genetic variation, all leading to genetic differentiation. For example, although they are the nearest populations geographically, genetic differentiation between Stony Creek and Dismal Creek was greatest ($F_{ST}$ = 0.95). Effects of this recent differentiation on candy darter fitness (i.e., survival, growth, reproduction) remain unexamined but could be important in the context of planning translocations to facilitate population recovery. In April 1996 (~9–12 generations ago), 30 adult candy darters were translocated from Stony Creek to Dismal Creek, as it was presumed that the population in Dismal Creek had been extirpated [14]. The occurrence of distinct and independent haplotypes with none shared between the two populations suggests that the translocation did not result in genetic augmentation of the native Dismal Creek population.

Within the phylogenetic reconstruction (Figure S1), one Kanawha darter sequence clustered with candy darter sequences. Kanawha darter is a sister lineage to candy darter, and its geographic range is just upstream within the New River drainage. With this as context, there are four plausible explanations for this unexpected phylogenetic result: (1) errant species identification in the field, (2) past hybridization between the species, such that Kanawha darter mtDNA was introgressed into candy darter background, (3) cyt *b* is under sufficient stabilizing selection pressure to make this interspecific comparison difficult or messy, and (4) the limited length of the mitochondrial sequence examined impaired distinction between the species. Competitive historical adjustment of their complementary ranges is indicated by the occurrence of the Little River population of *E. kanawhae* within the range of *E. osburni* [38]; the possibility that the species co-occurred in the past is supported by our observation of apparent mitochondrial introgression. Darters of the genus *Etheostoma* are known to hybridize [12,39,40], including candy and variegate darters. We reject the first explanation because of our extensive field experience (>15 person-years, collectively) sampling and handling candy darters. The latter three explanations can be tested by evaluating molecular genetic variation of more individuals at the zone of contact between the species and by screening of additional mitochondrial and nuclear DNA markers.

### 4.2. Contemporary Population Genetics of Candy Darter in the Upper New River

Each of the four candy darter populations exhibited departures of genotype frequencies from Hardy-Weinberg expectations, showing greater than expected homozygosity and indicating inbreeding within these small, isolated populations. The metric $F_{IS}$ quantifies the overall deficit of heterozygotes within populations. Although 95% confidence intervals for Inest estimates included zero, point values for $F_{IS}$ for all populations were positive, are likely the result of family structure within the populations, and may reflect some level of inbreeding. These departures could not be due to the Wahlund effect–the mixing of differentiated populations within the respective streams–because through the course of their life cycle, individuals within the respective populations use the full length of their stream [10]. Additionally, the G-W indices (*m*-ratios) indicate that each population has undergone a recent genetic bottleneck, sufficiently recent that lost microsatellite alleles have not yet re-arisen from new mutations.

Analyses of both mitochondrial and nuclear DNA markers revealed genetic differentiation among all populations. That AMOVA showed that most variance was within individuals is typical of vertebrates, although that component is often even larger; 24% of variance among populations is rather high [41–43]. All population pairwise comparisons revealed high levels of differentiation ($\geq$0.15 [44]). Dismal Creek and Stony Creek showed one of the highest levels of between-population differentiation, which supports the interpretation that the 1996 translocation failed to effectively supplement the Dismal Creek population. Results of STRUCTURE analysis also indicated a clear pattern of population genetic structuring among streams. A single individual in the Dismal Creek population (cluster number 2 in Figure 4) was assigned ancestry to all four populations; however, such an individual origin is highly unlikely given the stream network structure and the long fluvial distances between these streams. Collectively, these results support the interpretation that Virginia's contemporary candy darter populations are demographically and genetically isolated.

Estimates of $N_e$ based on a limited number of nuclear markers should be taken with caution. Using simulated data, Waples and Do [45] evaluated how use of highly polymorphic markers affects precision and bias in the single-sample method based upon linkage disequilibrium. They found that use of datasets with 10–20 loci with 10 alleles per locus and sample size of 50 yielded reasonably precise estimates of $N_e$ for small populations ($N_e < 200$) and that small populations were not likely to be mistaken for larger ones. While we screened 12 variable loci with sample sizes of 50–98, we observed 3.92–7.00 alleles per locus, with uncertain effects upon the precision of our $N_e$ estimates. Our estimates of effective population sizes indicated that all populations, except Dismal Creek ($N_e = 39$), had >50 breeders. An $N_e > 100$ is suggested to minimize risk of inbreeding depression and maintain short-term population viability, while an $N_e > 1000$ is needed for maintaining long-term evolutionary potential [46]. There is evidence of inbreeding and recent genetic bottlenecks in all populations, and confidence intervals around the point-estimates for $N_e$ are broad. Although $N_e$ estimates for the Cripple Creek, Laurel Creek, and Stony Creek populations are >100, these populations are still at risk of losing genetic variation and hence adaptive potential. This risk could become more evident over the long term if populations decline in the face of changing ecological conditions.

Genetic differentiation at both mitochondrial and microsatellite markers among Virginia populations of candy darter parallels that observed in other small, stream-dwelling percids. Similar levels of differentiation were found among isolated populations of Kentucky arrow darter *Etheostoma sagitta spilotum*, which collectively exhibited an isolation-by-distance [47] pattern of differentiation [48]; however, not all populations conformed to this pattern, suggesting that habitat separating populations may not permit dispersal. Although we observed no relationship between fluvial distance and differentiation of candy darter populations, there was a relationship between differentiation and position in the stream network relative to the New River. The least differentiation was observed between the two primary tributaries of the river, Stony Creek and Cripple Creek, which

are separated by 129 rkm. Such dispersal distances are not implausible for darters, especially when considering the potential combined movements by larval, juvenile, and adult life stages. For example, our range-wide analysis of variegate darter population genetic structure [49] showed that some populations were basically panmictic across much larger distances. Although their life-histories do not match perfectly, we know that juvenile Roanoke logperch *Percina rex* can disperse least 55 km [50]. In addition, Dismal Creek is the furthest removed (a tertiary tributary) from the New River and its population was the most differentiated from the others, based on pairwise comparisons of microsatellite data. The observed differentiation may have been exacerbated by population bottlenecks and inbreeding. Analysis of mitochondrial DNA revealed distinct haplotypes for each of the four populations, supporting the interpretation that genetic drift contributed to population genetic differentiation.

*4.3. Conservation Management Considerations*

Although planning to foster range-wide recovery of candy darter is coordinated by an interagency Candy Darter Species Recovery Team, management actions are taken within specific states and streams to mitigate specific threats. Virginia populations of candy darters face different threats (genetic isolation, with risks of inbreeding and loss of variation due to random genetic drift) than West Virginia populations (introgressive hybridization with the invasive variegate darter) [11,12] and this study. Our findings are especially germane to managing the genetic challenges facing Virginia's candy darters.

Loss of any population would significantly challenge long-term conservation of candy darter. Should the Virginia populations remain isolated, inbreeding and random genetic drift could diminish their demographic viability and adaptive potential, thereby limiting species recovery. A potential action to mitigate adverse effects of isolation is inter-stream translocation, which however may carry risk of outbreeding depression [48]. Given the evidence of historic connectivity, we expect effects from outbreeding depression to be less harmful to candy darter viability than any genetic rescue effects [4] that might result from outcrossing among the four Virginia populations. Assessments of the likelihood of outbreeding depression [51] may be needed to choose optimal source populations for translocating candy darter.

We suspect the observed differentiation among candy darter populations has resulted from genetic drift exacerbated by small $N_e$, and does not reflect local adaptation. Similar patterns have been reported for other imperiled fishes with fragmented populations. For example, Finger et al. [52] reached a similar conclusion regarding differentiation among remnant populations of Owens pupfish *Cyprinodon radiosus*. Pavlova et al. [53] demonstrated via simulations that responses by the endangered Macquarie perch *Macquaria australasica* to assisted gene flow would likely include increased genetic diversity and decreased probability of extinction and inbreeding. Similar outcomes are plausible for appropriately designed candy darter translocations. Our estimates of the genetically effective sizes of candy darter populations in Cripple, Laurel, and Stony creeks are well below the threshold recommended by Frankham et al. [46] to maintain evolutionary potential, but are above $N_e = 100$, the minimum recommended for use as source populations.

The inference of recent genetic bottlenecks in all Virginia populations indicates that all might benefit from genetic augmentation via translocations of wild or propagated fish. The Virginia Department of Wildlife Resources has expressed interest in translocating individuals from the lower 10 km of Cripple Creek to an unoccupied portion of upper Cripple Creek, approximately 5.6 km above the known distribution of candy darter (Michael Pinder, Virginia Department of Wildlife Resources, oral communication, 2021). This choice of source population would eliminate the possibility of outbreeding depression, as the source and new populations would share any local adaptations. An emphasis on translocations of wild fish could reduce the risk of a Ryman and Laikre [54] effect, wherein the genetic composition of a receiving population is overwhelmed by stocking the progeny of propagated individuals, which reduces the effective size of the wild population. We note that

translocations are already being conducted by a consortium of federal and state agencies aspiring to conserve candy darter in historically occupied streams below Bluestone Dam, West Virginia. Genetic analyses were conducted to ensure that the reintroduced individuals resembled the receiving population. The criteria used to guide these translocations, as well as their ultimate success or failure, could inform analogous choices regarding potential candy darter translocations in Virginia. There also may be lessons to learn from the apparent failure of the 1996 translocation of Stony Creek individuals into Dismal Creek. In particular, the site selected and the single release of fish may have been poor choices. This translocation site was below a natural barrier, Dismal Falls [15], which may have constrained upstream movement and subjected translocated individuals to competition with other darter species. Given the isolation of Virginia's candy darter populations, we suggest that any augmentations need to be conducted intermittently over the long term, as genetically effective natural dispersal between populations seems highly unlikely.

## 5. Conclusions

We examined genetic variation within and among the four isolated populations of candy darter that occur above Bluestone Dam, West Virginia, which protects them from invasion by and introgressive hybridization with the invasive variegate darter. While all four populations show evidence of isolation, random genetic drift, and inbreeding, two showed signatures of historical immigration from other, differentiated populations. Overall, our findings indicate that habitat management to allow spatial and demographic expansion of populations, and perhaps also translocations to exchange genetic material among populations, may be effective tactics to promote conservation of candy darter in Virginia.

**Supplementary Materials:** The following supporting information can be downloaded at: https://www.mdpi.com/article/10.3390/fishes8100490/s1, Table S1. Site locations for Cripple Creek, Dismal Creek, Stony Creek, and Laurel Creek for all surveys. Coordinates are in decimal degrees. Table S2. Microsatellite locus-by-locus metrics of genetic diversity for four populations of candy darter *Etheostoma osburni* in Virginia. Table S3. Mean log probabilities, $\text{Ln}P(D|K)$, supporting given numbers of multilocus microsatellite DNA genotypic clusters ($K$) for candy darter *Etheostoma osburni* provided by STRUCTURE Bayesian cluster analysis. Figure S1. Current and historical distribution of the candy darter, *Etheostoma osburni* (USFWS 2018). Green indicates extant Candy Darter populations; yellow indicates historical or extirpated populations. Red lines are major dams that present barriers to fish movement. Figure S2. Maximum-likelihood phylogenetic tree of cytochrome *b* sequences of four Virginia candy darter *Etheostoma osburni* populations. Figure S3. Mitochondrial cyt *b* DNA sequence mismatch distributions of pairwise nucleotide differences in the (a) Cripple Creek, (b) Dismal Creek, (c) Laurel Creek, and (d) Stony Creek populations of candy darter *Etheostoma osburni*. Figure S4. Results from application of the Evanno et al. (2005) method for determining the best-supported number of genetic clusters ($K$) using 12 microsatellite DNA loci for four collections of candy darter *Etheostoma osburni* in STRUCTURE.

**Author Contributions:** Conceptualization, P.L.A. and E.M.H.; funding acquisition, P.L.A. and E.M.H.; methodology, K.E.M., E.M.H. and P.L.A.; field and laboratory work, K.E.M.; data analysis, K.E.M., E.M.H. and P.L.A.; writing of original draft, K.E.M.; review and editing of manuscript, P.L.A. and E.M.H. All authors have read and agreed to the published version of the manuscript.

**Funding:** This research was supported by the Virginia Department of Wildlife Resources and Virginia Polytechnic Institute and State University.

**Institutional Review Board Statement:** This work was carried out under the auspices of the Virginia Tech Institutional Care and Use Committees Protocol 16-095, first approved 29 September 2016.

**Data Availability Statement:** At the time of publication, data were not publicly available from Virginia Department of Wildlife Resources. The data that support the findings of this study will be available on request to the corresponding author.

**Acknowledgments:** We thank Mike Pinder and Craig Roghair for their in-kind support in conducting fieldwork, and Logan Sleezer for developing the maps. The participation of author EMH was supported in part by the U.S. Department of Agriculture National Institute of Food and Agriculture Hatch Program. The Virginia Cooperative Fish and Wildlife Research Unit is jointly sponsored by U.S. Geological Survey, Virginia Tech, Virginia Department of Wildlife Resources, and Wildlife Management Institute. Any use of trade, firm, or product names is for descriptive purposes only and does not imply endorsement by the U.S. Government.

**Conflicts of Interest:** The authors declare no conflict of interest.

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
