# Peer review of "Genetic Structure across Isolated Virginia Populations of the Endangered Candy Darter (Etheostoma osburni)"

_fishes, doi:10.3390/fishes8100490_

Round 1

Reviewer 1 Report

The manuscript titled “Genetic structure across isolated Virginia populations of the Endangered Candy Darter”highlights the importance of understanding population genetic structure in endangered species conservation. The Candy Darter populations in Virginia have faced challenges related to genetic bottlenecks and isolation, which could impact their long-term viability. The findings support the implementation of habitat management strategies to promote population expansion and the exchange of genetic material among populations through translocations. Maintaining larger effective population sizes is critical for preserving the evolutionary potential of the Candy Darter. This study provides valuable insights for conservation management efforts aimed at ensuring the survival and recovery of this federally endangered fish species in Virginia. The study provides valuable scientific insights into the genetic structure and population dynamics of the endangered Candy Darter in Virginia. The research methods employed, including the analysis of mitochondrial DNA and microsatellite loci, contribute to the robustness of the findings. The identification of genetic bottlenecks and the genetic differentiation among populations have important implications for conservation strategies.

The research is novel and authors did a good deal of analytical work. I would love to recommend it to be accepted by Fishes journal, if authors get a chance to major revise it addressed my specific comments. 

1. The reference citation format is peculiar and needs to be modified according to the journal's requirements.

2. Table 6: The numbers below "n" should not be italicized.

3. During sampling, it is important to determine if the samples belong to the same species. Considering the significant differentiation observed between sampling points, it raises the question of whether other species are involved. Has there been any morphological or molecular biological identification conducted?

4. Figure 2: It is suggested to combine all sampling points into one comprehensive image instead of the current arrangement, particularly as the "E" part differs from the others.

5. Line 276, Figure 3: Why is it italicized?

6. Figure 3: How was "E.kan" included in the graph? Typically, in such studies, outgroups are not added, and the haplotype network is constructed for the same species only.

7. Considering the current Figure 3, how can we explain the connection between the two haplotype networks of the target species with another species? Does this imply that these two blocks belong to different species? This intensifies my suspicion, considering the points raised in the third comment and also the results in Figure 4.

Author Response

Response to Reviewer 1

We have received reviews of our manuscript, “Genetic structure across isolated Virginia populations of the Endangered Candy Darter” - ID: Fishes-2540908. We have attended to the comments of the reviewers to yield a revised manuscript, which we resubmit in two formats, showing Tracked Changes and “clean”. Line numbers below refer to the line numbers in the Tracked Changes version. We have deleted authors’ names from the in-text literature citations in accordance with journal stylistics.

Reviewer 1

The manuscript titled “Genetic structure across isolated Virginia populations of the Endangered Candy Darter” highlights the importance of understanding population genetic structure in endangered species conservation. The Candy Darter populations in Virginia have faced challenges related to genetic bottlenecks and isolation, which could impact their long-term viability. The findings support the implementation of habitat management strategies to promote population expansion and the exchange of genetic material among populations through translocations. Maintaining larger effective population sizes is critical for preserving the evolutionary potential of the Candy Darter. This study provides valuable insights for conservation management efforts aimed at ensuring the survival and recovery of this federally endangered fish species in Virginia. The study provides valuable scientific insights into the genetic structure and population dynamics of the endangered Candy Darter in Virginia. The research methods employed, including the analysis of mitochondrial DNA and microsatellite loci, contribute to the robustness of the findings. The identification of genetic bottlenecks and the genetic differentiation among populations have important implications for conservation strategies.

The research is novel and authors did a good deal of analytical work. I would love to recommend it to be accepted by Fishes journal, if authors get a chance to major revise it addressed my specific comments.

  1. The reference citation format is peculiar and needs to be modified according to the journal's requirements.

RESPONSE: The journal uses a numbered references format; knowing that we faced inevitable revisions and that it is hard to track the correspondence of numbers and authors, for the review draft, we presented both numbers and authors. In this revised draft, we present only the numbers, as the journal requires.

  1. Table 6: The numbers below "n" should not be italicized.

RESPONSE: Reviewer 1 caught an error, which we have fixed.

  1. During sampling, it is important to determine if the samples belong to the same species. Considering the significant differentiation observed between sampling points, it raises the question of whether other species are involved. Has there been any morphological or molecular biological identification conducted?

RESPONSE: Coauthor Angermeier has been working with candy darters for over 12 years. Author McBaine was trained to identify candy darter and co-occurring species before conducting any sampling. The only species with which it might be confused is Kanawha darter, whose current range is disjunct from that of candy darter. We sequenced the mitochondrial cytochrome b gene of our fish, and the sequences aligned with those of candy darter. In short, we are very confident that the fish that we call “candy darter” are indeed candy darter.

  1. Figure 2: It is suggested to combine all sampling points into one comprehensive image instead of the current arrangement, particularly as the "E" part differs from the others.

RESPONSE: We have deleted the old map and added a new one showing all riverways in the study area. We have deleted the four small panels. We show sampling locations as dots. We have added the larger regional map as an inset into the map.

  1. Line 276, Figure 3: Why is it italicized?

RESPONSE: That was a simple error, it should have been in bold font. It has been fixed.

  1. Figure 3: How was "E.kan" included in the graph? Typically, in such studies, outgroups are not added, and the haplotype network is constructed for the same species only.

RESPONSE: There are instances where authors do include outgroups as we have. When we did the haplotype analysis with the outgroup, this led to the interesting result that there is evidence of an ancient hybridization event between the species. This is an interesting finding, and hence, it is appropriate to include this result in the haplotype network. We do discuss this in the Discussion at line 449-460.

  1. Considering the current Figure 3, how can we explain the connection between the two haplotype networks of the target species with another species? Does this imply that these two blocks belong to different species? This intensifies my suspicion, considering the points raised in the third comment and also the results in Figure 4.

RESPONSE: Again, the respective species had an ancient hybridization event, as we explained in the discussion. The individuals that we assayed identified phenotypically as candy darters. The respective species now have disjunct distributions and are not known to hybridize.

As for Figure 4, the populations share many microsatellite alleles, but at different frequencies. Hence, the distinct clustering.

Summary comment

We hope that with these revisions, our manuscript is found suitable for publication in Fishes.

                                                Regards,

                                                Eric Hallerman

Reviewer 2 Report

Interesting paper, solid results, acceptable after minor revision.

ABSTRACT

  • Here and all thru the ms, species names are not capitalised. So the name of your species should be written as “candy darter”. Same for “endangered”, why did you write Endangered all thru the ms? P
  • In this regard, also thru the ms, change “imperilled” to “endangered”. Nobody used the term “imperilled species”. 
  • I find the abstract very unbalanced. The aims section is too long, no need to specify and number all your aims in the abstract. On the other hand it lacks specific details about the M&M, how many individuals you analysed, from how many locations, etc. And also lacks results. There are barely any results.

INTRODUCTION

  • Line 37. I find the start quite weak. Specially starting with a sentence like “Declining species”, what are “declining species”, it’s a not a good start. Also the whole first paragraph reads a bit outdated.
  • Line 51. Define what is genetic augmentation.
  • Line 61. You mention that the life history/fecundity of the species has been studied and you cite Reference 9 but say nothing else. Then you describe life history/fecundity in detail and cite something else, Reference 10. We all like to get more citations but this is a bit ridiculous. 
  • Line 69. In the paragraph, be careful with the use of the term “populations”. Locations where we sample and populations are not the same thing. Also careful with the use of “samples” and “individuals” thru the ms.

MATERIAL & METHODS

  • I cannot find any information in the M&M about how many individuals you sampled at which location, years, dates etc. You need a table with that info. I can deduce it from a later table. But in the Fields section or in a new section in M&M, you have to explain number of samples per location, year in which they were sampled in each location etc and that has to be in a Table too. This has to be in M&M, not in Results. Also from one of the supplementary tables, I get the feeling there are samples from different sections without the rivers, that they are not all from the same place, you also have to detail all this.  
  • Figure 2 with the sampling needs to be improved 1000% and also combined with other info to make it more interesting. So you need a big map, similar like the map in the Supplementary material. If the map is bigger then there is no need with the 4 small panels (A-D), remove those. Also include all the rivers in the region, like in the map in the supplementary material, which is actually a much better figure than this one. So the map has to have all the region, with all rivers, and the ones you studied highlighted. Then also add the sampling stations, with like dots. And also panel E should be included very small inside the main map, as a sub-panel within the big map.
  • Then in order to make the map more interesting, add some genetic data. For instance, you can make pies and show the proportion of haplotypes for each of the 4 locations. That’s a classic figure that everybody does and it is much easier to visualise the data from Table 1 this way. So basically take the data from Table 1, make pies, and include them in Figure 2 with the sampling.
  • I’m very skeptical about the use of 12 Microsatellites to do Ne estimations. This is no longer 1985, people do these things now with genomic approaches, nobody does that with microsatellites and especially with only 12. So include it here if you want, but add a statement about the little reliability of the method and that those Ne calculations should be taken with a pinch of salt.
  • Again, we are in 2023. To do demographic analyses doing mismatch distribution is obsolete. It was already dubious and circumstancial 40 years ago but nowadays with all the genome approaches around, this analysis makes no sense, so I suggest removing it. You have a strong dataset and you have done a good job analysing it. But you have the markers to do basic genetic data analysis, so do FSTs and do haplotype analysis. But trying to calculate Ne and then doing demographic analysis based on the one marker is kinda over-reaching and ruining the manuscript. Your markers are just not good enough to do so. I do suggest removing all the mismatch distribution analysis.

RESULTS

  • Table 1. Are haplotypes 1 and 2 the same? I see no differences!

  • Figure 3. Don’t understand the Figure. Why is Hap 3 on top of Hap 2? There should be 1 difference/change away. 
  • Moreover in Figure 3, the circles are not proportional. Did you use the software? Or did you do it yourself manually? Cos it seems the later. The circles with 47 individuals should be much bigger.
  • Line 283. Have you talked about Mismatch analysis before? Why you suddenly mention it here in Results if it’ not in M&M? Also the sentence makes no sense (“Were only mutation and random genetic drift at issue…”), what does it even mean?

DISCUSSION
The Discussion needs more proper discussion and less repetition of results. In each section, 80% is a repeatition of the results that we already know. You have to summarise the results in 1 line and then discuss proper. Repeating the results is not discussing, we already read the results section. 

Overall, careful with over-interpreting the results from the mitochondrial data. That’s just 1 marker. There are million of markers in the genome.

Also you need to discuss more the fact that the current Dismal Creek pop originates from 30 individuals from Stony Creek. This is a very interesting fact and you only partially discuss it. So in 20 years the populations have become completely different . This should be addressed and 1000% properly discussed. In 20 years 2 pops can get an FST of 0.3. That’s more interesting than 90% of what you’re currently discussing in the Discussion. 

As mentioned above, please add something pointing out the limitations of estimating Ne on the basis of 12 microsatellites and that in 2023 nobody does this anymore and we have better ways available thanks to the genomic revolution. Is it ok to include the Ne results but add the limitation of the method used.

CONCLUSIONS

Delete. You already repeated the same things twice in Results and Discussion, no need for a third time.

See above

Author Response

Response to Reviewer 2

We have received reviews of our manuscript, “Genetic structure across isolated Virginia populations of the Endangered Candy Darter” - ID: Fishes-2540908. We have attended to the comments of the reviewers to yield a revised manuscript, which we resubmit in two formats, showing Tracked Changes and “clean”. Line numbers below refer to the line numbers in the Tracked Changes version. We have deleted authors’ names from the in-text literature citations in accordance with journal stylistics.

Reviewer 2

Interesting paper, solid results, acceptable after minor revision.

RESPONSE: We are pleased with the positive overall comment.

ABSTRACT

Here and all thru the ms, species names are not capitalised. So the name of your species should be written as “candy darter”. Same for “endangered”, why did you write Endangered all thru the ms? P

In this regard, also thru the ms, change “imperiled” to “endangered”. Nobody used the term “imperilled species”.

RESPONSE: We have decapitalized “candy darter” throughout the manuscript. “Endangered” denotes a legal protection status and is routinely presented in capitalized form in many conservation-related documents – hence, our use of capitalization in the draft manuscript. “Imperiled” denotes conservation concern without regard to status of legal protection. Despite the reviewer’s comment, It is widely used in the conservation literature.  

I find the abstract very unbalanced. The aims section is too long, no need to specify and number all your aims in the abstract. On the other hand it lacks specific details about the M&M, how many individuals you analysed, from how many locations, etc. And also lacks results. There are barely any results.

RESPONSE: We have reworked the Abstract to achieve better balance of aims, methods and results. We have removed numbers from the statement of objectives.

INTRODUCTION

Line 37. I find the start quite weak. Specially starting with a sentence like “Declining species”, what are “declining species”, it’s a not a good start. Also the whole first paragraph reads a bit outdated.

RESPONSE: We define “declining species” in the first three sentences of the Introduction. We disagree with the reviewer’s characterization of the first paragraph as not a good start. It lays out why a general reader should be interested in the species of interest, i.e., that our findings will have applicability to a wider range of aquatic species of conservation interest. The paragraph leads to three sentences that explain how our findings can prove useful for conservation-oriented management. 

Line 51. Define what is genetic augmentation.

RESPONSE: We have clarified the intent of the sentence by referring to populations that “may need augmentation with new genetic variation via stocking.”

Line 61. You mention that the life history/fecundity of the species has been studied and you cite Reference 9 but say nothing else. Then you describe life history/fecundity in detail and cite something else, Reference 10. We all like to get more citations but this is a bit ridiculous.

RESPONSE: We have revised the passage at issue to address the comment. We make clearer what each supporting study contributes. The passage now reads: “Its abundance, age structure, and life history in the four Virginia streams in which it occurs recently have been described. The species has greatest abundance in Stony and Cripple Creeks [McBaine and Hallerman 2020 = 9]. Individuals can live to at least age-5, based on otolith readings. Females can mature at age-2, then reproduce in four consecutive years with multiple males each year; parentage analysis indicates that both sexes mate with multiple partners [McBaine et al. 2022 = 10].” Excessive self-citation is not at issue; both citations make their unique contributions.

Line 69. In the paragraph, be careful with the use of the term “populations”. Locations where we sample and populations are not the same thing. Also careful with the use of “samples” and “individuals” thru the ms.

RESPONSE: The reviewer is correct – locations of occurrence and genetic populations are not the same. We have revised the paragraph in question. Because we do not know the population genetic structure of candy darter in its originally known 35 locations, we now refer to those as locations of occurrence as opposed to populations. 

MATERIAL & METHODS

I cannot find any information in the M&M about how many individuals you sampled at which location, years, dates etc. You need a table with that info. I can deduce it from a later table. But in the Fields section or in a new section in M&M, you have to explain number of samples per location, year in which they were sampled in each location etc and that has to be in a Table too. This has to be in M&M, not in Results. Also from one of the supplementary tables, I get the feeling there are samples from different sections without the rivers, that they are not all from the same place, you also have to detail all this. 

RESPONSE: We have added a new Table S1 that lays out locations and years in detail. We have added numbers of samples subjected to mitochondrial analysis at line 266 and to microsatellite analysis at lines 334-335.  

Figure 2 with the sampling needs to be improved 1000% and also combined with other info to make it more interesting. So you need a big map, similar like the map in the Supplementary material. If the map is bigger then there is no need with the 4 small panels (A-D), remove those. Also include all the rivers in the region, like in the map in the supplementary material, which is actually a much better figure than this one. So the map has to have all the region, with all rivers, and the ones you studied highlighted. Then also add the sampling stations, with like dots. And also panel E should be included very small inside the main map, as a sub-panel within the big map.

RESPONSE: We have deleted the old map and added a new one showing all riverways in the study area. We have deleted the four small panels. We show sampling locations as dots. We have added the larger regional map as an inset into the map. In Table S1, we present latitude-longitude coordinates to only ne significant digit as the U.S. Fish and Wildlife Service discourages publishing specific locations of endangered spp. Candy darters are at especially high risk, as some may want to collect them for use as bait or aquaria.  

Then in order to make the map more interesting, add some genetic data. For instance, you can make pies and show the proportion of haplotypes for each of the 4 locations. That’s a classic figure that everybody does and it is much easier to visualise the data from Table 1 this way. So basically take the data from Table 1, make pies, and include them in Figure 2 with the sampling.

RESPONSE: We feel that it is just too much to add genetics data to this map of the study area. It makes the figure too busy, and the geographic distribution of mitochondrial haplotypes is well depicted in Figure 3.

I’m very skeptical about the use of 12 Microsatellites to do Ne estimations. This is no longer 1985, people do these things now with genomic approaches, nobody does that with microsatellites and especially with only 12. So include it here if you want, but add a statement about the little reliability of the method and that those Ne calculations should be taken with a pinch of salt.

RESPONSE: Several points are relevant here:

First, microsatellite DNA markers were not widely utilized until about 1995, fully ten years after the date mentioned above.

Second, SNPs have indeed become the go-to tool for well-studied species such as salmonids for which there are reference genomic sequences. Such is not the case for non-game and imperiled species, for which microsatellites are still widely used. As a peer reviewer and journal editor, I handle about 100 fisheries genetics papers per year. Once you move away from salmonids and important aquaculture species, half or more of all such manuscripts involve use of microsatellites, often with about ten loci per study. This choice of experimental approaches is driven largely by the lack of research funding to support development of the reference genome data needed to apply SNPs-based studies rigorously. Third, we have added a passage to the Discussion at lines 490-487 regarding caveats for use of microsatellite data to estimate Ne. It reads: Estimates of Ne based on a limited number of nuclear markers should be taken with caution. Using simulated data, Waples and Do [36] evaluated how use of highly polymorphic markers affects precision and bias in the single-sample method based upon linkage disequilibrium. They found that use of datasets with 10-20 loci with 10 alleles per locus and sample size of 50 yielded reasonably precise estimates of Ne for small populations (Ne <200) and that small populations were not likely to be mistaken for larger ones. While we screened 12 variable loci with sample sizes of 50-98, we observed 3.92-7.00 alleles per locus, with uncertain effects upon the precision of our Ne estimates.  

Again, we are in 2023. To do demographic analyses doing mismatch distribution is obsolete. It was already dubious and circumstancial 40 years ago but nowadays with all the genome approaches around, this analysis makes no sense, so I suggest removing it. You have a strong dataset and you have done a good job analysing it. But you have the markers to do basic genetic data analysis, so do FSTs and do haplotype analysis. But trying to calculate Ne and then doing demographic analysis based on the one marker is kinda over-reaching and ruining the manuscript. Your markers are just not good enough to do so. I do suggest removing all the mismatch distribution analysis.

RESPONSE: We disagree. We performed a quick search of Google Scholar on the term “sequence mismatch distribution”, restricting the search to 2023 publications. Noting only papers on fishes and aquatic or marine invertebrates, the search produced publications in the leading journal Gene, as well as four fisheries and several other journals:

Cui X, Yang M, Li C, An B, Mu S, Zhang H, Chen Y, Li X, Kang X. Assessment of genetic diversity and population structure of Neocaridina denticulata sinensis in the Baiyangdian drainage area, China, using microsatellite markers and mitochondrial cox1 gene sequences. Gene. 2023 Jun 5:147534.

Zi-Ping YA, Da-Ming LI, Yan-Shan LI, Jia-Xin YA. Genetic diversity and genetic structure of Culter alburnus in Tai Lake and Hongze Lake based on mitochondrial DNA Cyt b gene sequences. Journal of Fisheries Research. 2023 Feb 25;45(1):1.

Rumisha C, Kochzius M. Genetic evidence for a single stock of giant tiger prawns Penaeus monodon in demarcated prawn fishing zones of Tanzania. Fisheries Management and Ecology. 2023 Feb;30(1):36-43.

Luo Y, Zhang Y, Cheng R, Li Q, Zhang Y, Li Y, Shen Y. Genetic Diversity of Jinshaia sinensis (Cypriniformes, Balitoridae) Distributed Upstream of the Yangtze River. Fishes. 2023 Jan 28;8(2):75.

Boquiren AR, Quilang JP. Genetic diversity and population structure of the crescent perch Terapon jarbua (Centrarchiformes: Terapontidae) in the Philippines. Regional Studies in Marine Science. 2023 Feb 1;58:102786.

Dai X, Tian S, Wu W, Yang B, Ma Y, Ge G, Wu D. Phylogeography of alpine plant Parnassia wightiana (Celastraceae). Botany. 2023 Jun 14(ja).

Ueki G, Tojo K. The phylogeography of the stag beetle Dorcus montivagus (Coleoptera, Lucanidae): Comparison with the phylogeography of its specific host tree, the Japanese beech Fagus crenata. Entomological Science. 2023 Mar;26(1):e12535.

Zhang HC, Hu TG, Shi CY, Chen GW, Liu DZ. Population Analysis Based on Mito-nuclear Sequences: Implication of Dugesia japonica Decline in Taihang Mountains, China. Pakistan Journal of Zoology. 2023 Apr 1;55(2).

Clearly, sequence mismatch analysis is not obsolete. We applied this analytic approach because fundamentally, how better to detect the secondary contact of once-isolated populations that bore different mitochondrial lineages?

RESULTS

Table 1. Are haplotypes 1 and 2 the same? I see no differences!

RESPONSE: The reviewer caught an error for which we are grateful. We have added a corrected Table 1.

Figure 3. Don’t understand the Figure. Why is Hap 3 on top of Hap 2? There should be 1 difference/change away. Moreover in Figure 3, the circles are not proportional. Did you use the software? Or did you do it yourself manually? Cos it seems the later. The circles with 47 individuals should be much bigger.

RESPONSE: This issue stemmed from the error caught in the comment just above. We did use Popart to create the haplotype network, and find that the circles are only roughly proportional to the numbers of the respective haplotypes. We have changed the figure and its caption to correct the issues noted by the reviewer, for which we are grateful.

Line 283. Have you talked about Mismatch analysis before? Why you suddenly mention it here in Results if it’ not in M&M?

RESPONSE: At lines 189-191, we had written: “In addition, we assessed pairwise mitochondrial DNA sequence mismatch distributions within populations as an indication of historical demography conducted using DnaSP.” The reviewer apparently missed that.

Also the sentence makes no sense (“Were only mutation and random genetic drift at issue…”), what does it even mean?

RESPONSE: At line 295, a bit of wordsmithing yields a clearer sentence, starting as “If only mutation and random genetic drift had been at issue”.

DISCUSSION

The Discussion needs more proper discussion and less repetition of results. In each section, 80% is a repeatition of the results that we already know. You have to summarise the results in 1 line and then discuss proper. Repeating the results is not discussing, we already read the results section.

RESPONSE: We have reworked and sharpened the Discussion. Some repetition of results to set up interpretive passages in the discussion is unavoidable. Additionally, modest repetition is need to set up separate discussions of historical demography, contemporary population genetics, and implications for management. We feel that revision has yielded a tight Discussion section.    

Overall, careful with over-interpreting the results from the mitochondrial data. That’s just 1 marker. There are million of markers in the genome.

RESPONSE: The reviewer is correct that mitochondrial DNA provides but one marker – hence our sequencing of just one mitochondrial DNA region. Yet, we chose to include mitochondrial DNA in our study because its variation is useful for reconstructing recent phylogeny and historical demography. 

We have striven not to over-interpret. We add that it would take scores of microsatellite loci or hundreds to thousands of SNP loci to attempt such reconstructions.  

Also you need to discuss more the fact that the current Dismal Creek pop originates from 30 individuals from Stony Creek. This is a very interesting fact and you only partially discuss it. So in 20 years the populations have become completely different . This should be addressed and 1000% properly discussed. In 20 years 2 pops can get an FST of 0.3. That’s more interesting than 90% of what you’re currently discussing in the Discussion.

RESPONSE: The Dismal Creek population is a native population, and we have no idea how the reviewer got that idea that the Dismal Creek population originated from stocking from Stony Creek. Hence, the rest of this comment has no basis.

As mentioned above, please add something pointing out the limitations of estimating Ne on the basis of 12 microsatellites and that in 2023 nobody does this anymore and we have better ways available thanks to the genomic revolution. Is it ok to include the Ne results but add the limitation of the method used.

RESPONSE: As noted above, we have added a passage to the Discussion at lines 490-4897 to acknowledge caveats about use of microsatellite loci to estimate Ne. Again, many researchers still apply the approach, especially those working with species of conservation interest and limited funding. 

CONCLUSIONS

Delete. You already repeated the same things twice in Results and Discussion, no need for a third time.

RESPONSE: The journal specifically asks for a Conclusions section. Hence, we provided one. 

Summary comment

We hope that with these revisions, our manuscript is found suitable for publication in Fishes.

                                                Regards,

                                                Eric Hallerman

Round 2

Reviewer 1 Report

Modified as required and agreed to accept

Author Response

Response to Academic Editor

For the Points 3, 6, and 7 of Reviewer 1, the authors may provide another evidence, such as phylogenetic analysis of nuclear gene, to make this interested finding clear.

RESPONSE: We have pasted in these comments from Reviewer 1 and provide complete responses below.

  1. During sampling, it is important to determine if the samples belong to the same species. Considering the significant differentiation observed between sampling points, it raises the question of whether other species are involved. Has there been any morphological or molecular biological identification conducted?

RESPONSE: Kanawha darter Etheostoma kanawhae and candy darter E. osburni form a species pair that is endemic to the New River drainage (Jenkins and Burkhead 1994). E. kanawhae is endemic to the upper and middle New River drainage of North Carolina and Virginia, where it is virtually restricted to the Blue Ridge geographical province. E. osburni occurs in the middle and lower New River drainage of Virginia and West Virginia in the Ridge and Valley and Cumberland Plateau provinces. Both species probably arose in their respective, disjunct distributions from an E. variatum-like ancestor (Jenkins and Burkhead 1994). Their intimate phylogenetic relationship is shown by strong similarities in coloration, morphology, and life histories.

  1. kanawhae is distinguished from E. osburni (Jenkins and Burkhead 1994) by its larger scales: (1) lateral line scales [47(50-58) in kanawhae versus (58)59-66(71) in osburni]; (2) scales above the lateral line [7-6(8) versus (7)8-9]; (3) scales below the lateral line [7-8(10) versus 9-11]; and circumpeduncle scales [18-21 versus (22)23-24(26)]. Also differing is the location of the posterior-most pore of the anterior section of the infraorbital canal, generally pore 4. In kanawhae, it occurs at the end of a short- or medium-length canaliculus that extends upward, or occasionally it open on the main canal or on a short, ventrally-extending canaliculus. In osburni, the pore occurs at the end of a long canaliculus that extends upwards almost to the eye. Life colors of males differ on average, but this may relate to fish size. Compared with kanawhae, larger osburni are more ornate, with the cheek orange-red and small additional small red marks present on head and breast; blue-green more widespread on the body; more bars present on body; red bars on body larger, more pronounced, most or all of the anterior bars connected to red flank stripe; most red bars set off from blue-green by a white border; 1-2 red spots develop within certain blue-green bars in some males; all fins except the first dorsal have more red; and clear band in first dorsal fin narrowed or obliterated by a wider medial bluish band.
  2. Figure 3: How was "E.kan" included in the graph? Typically, in such studies, outgroups are not added, and the haplotype network is constructed for the same species only.

RESPONSE: There are instances where authors do include outgroups as we have. When we did the haplotype analysis with the outgroup, this led to the interesting result that there is evidence of an ancient hybridization event between the species. Jenkins and Burkhead (1994) wrote that competitive historical adjustment of their complementary ranges is indicated by the occurrence of the Little River population of E. kanawhae within the range of E. osburni; the possibility that they co-occurred in the past is supported by our observation of apparent mitochondrial introgression. This is an interesting finding, and hence, we find it appropriate to include this result in the haplotype network. We now mention this in the Discussion at lines 438-443.

  1. Considering the current Figure 3, how can we explain the connection between the two haplotype networks of the target species with another species? Does this imply that these two blocks belong to different species? This intensifies my suspicion, considering the points raised in the third comment and also the results in Figure 4.

RESPONSE: While the species do differ at mitochondrial sequences (our results; see also www.boldsystems.org), nuclear DNA differentiation among the species has not been characterized. We do not have grant support to execute that work at this time. We note that darters of the genus Etheostoma are known to hybridize (examples have been shown by Ray et al. 2008, Bossu and Near 2013, and Gibson et al. 2019, the latter involving candy and variegate darters). We infer that candy and Kanawha darter had an ancient hybridization event, as we explain in the discussion at lines 438-443. The individuals that we assayed were identified phenotypically as candy darters. The respective species now have disjunct distributions and are not known to hybridize. We choose to report our interesting finding regarding historical hybridization for those building upon our work to see.

Bossu, C.M.; Near, T.J. Characterization of a contemporaneous hybrid zone between two darter species (Etheostoma bison and E. caeruleum) in the Buffalo River System. Genetica 2013, 141, 75-88.

Gibson, I.; Welsh, A.B.; Welsh, S.A.; Cincotta, D.A. Genetic swamping and possible species collapse: Tracking introgression between the native candy darter and introduced variegate darter. Conserv. Genet. 2019, 20, 287-298.

Jenkins, R.E.; and Burkhead, N.M. Freshwater Fishes of Virginia. American Fisheries Society, Bethesda, MD, 1994.

Ray, J.M.; Lang, N.J.; Wood, R.M.; Mayden, R.L. History repeated: recent and historical mitochondrial introgression between the current darter Etheostoma uniporum and rainbow darter Etheostoma caeruleum (Teleostei: Percidae). J Fish Biol 2008, 72(2), 418-434.